# Quantifying the impact of taking medicines for primary prevention: a time-trade off study to elicit direct treatment disutility in the UK

Alexander Thompson ![ORCID],[1] Ji-Hee Youn,[1] Bruce Guthrie ![ORCID],[2,3] Robert Hainsworth ![ORCID],[1] Peter Donnan ![ORCID],[4] Gabriel Rogers ![ORCID],[1] Daniel Morales,[5] Katherine Payne ![ORCID] [1]

[1]Manchester Centre for Health Economics, The University of Manchester, Manchester, UK
[2]Advanced Care Research Centre, University of Edinburgh, Edinburgh, UK
[3]Usher Institute, College of Medicine and Veterinary Medicine, University of Edinburgh, Edinburgh, UK
[4]Dundee Epidemiology and Biostatistics Unit, University of Dundee, Dundee, UK
[5]Division of Population Health Sciences, University of Dundee, Dundee, UK

**Correspondence to**
Dr Alexander Thompson;
alexander.thompson@
manchester.ac.uk

## ABSTRACT

**Background** Direct treatment disutility (DTD) represents an individual's disutility associated with the inconvenience of taking medicine over a long period of time.

**Objectives** The main aim of this study was to elicit DTD values for taking a statin or a bisphosphonate for primary prevention. A secondary aim was to understand factors which influence DTD values.

## Methods

**Design:** We used a cross-sectional study consisting of time-trade off exercises embedded within online surveys. Respondents were asked to compare a one-off pill ('Medicine A') assumed to have no inconvenience and a daily pill ('Medicine B') over 10 years (statins) or 5 years (bisphosphonates).

**Setting:** Individuals from National Health Service (NHS) primary care and the general population were surveyed using an online panel company.

**Participants:** Two types of participants were recruited. First, a purposive sample of patients with experience of taking a statin (n=260) or bisphosphonate (n=100) were recruited from an NHS sampling frame. Patients needed to be aged over 30, have experience of taking the medicine of interest and have no diagnosis of dementia or of using dementia drugs. Second, a demographically balanced sample of members of the public were recruited for statins (n=376) and bisphosphonates (n=359).

**Primary and secondary outcome measures:** Primary outcome was mean DTD. Regression analysis explored factors which could influence DTD values.

**Results** A total of 879 respondents were included for analysis (514 for statins and 365 for bisphosphonates). The majority of respondents reported a disutility associated with medicine use. Mean DTD for statins was 0.034 and for bisphosphonates 0.067, respectively. Respondent characteristics including age and sex did not influence DTD. Experience of bisphosphonate-use reduced reported disutilities.

**Conclusions** Statins and bisphosphonates have a quantifiable DTD. The size of estimated disutilities suggest they are likely to be important for cost-effectiveness, particularly in individuals at low-risk when treated for primary prevention.

## STRENGTHS AND LIMITATIONS OF THIS STUDY

⇒ We use a large representative sample of both public and patients for our estimated values.
⇒ We used patients to inform and trial our survey and designed storyboards and visual arrays to contextualise the exercises and communicate absolute risk.
⇒ The time trade-off methods employed were consistent with those first used to derive the EQ-5D-3L, a widely used measure of health-related quality of life.
⇒ Survey respondents self-selected and this could be related to their ability to complete the survey or their self-reported health.

## INTRODUCTION

Medicines for the primary prevention of disease are typically taken long-term, often for life, following identification that an individual is at risk of a harmful event, such as a stroke due to cardiovascular disease[1] or a hip fracture as a result of osteoporosis.[2] Taking a medicine is considered, in general, to have a minimal day-to-day treatment burden because, in the traditional clinical sense, the action of medicine-taking is perceived to be non-invasive. Yet there is qualitative evidence to suggest that taking a medicine long-term is associated with treatment burden.[3] Moreover, supporting this qualitative evidence is a quantitative literature seeking to estimate the utility of pill-taking, or alternatively, the negative impact of undergoing long-term treatment with medicines, called 'direct treatment disutility'.

Direct treatment disutility (DTD) represents an individual's strength of preference not to take a medicine. DTD could occur for a number of reasons including the inconvenience of obtaining prescriptions and medicines, needing to modify lifestyles to take medicines and attending healthcare

visits for monitoring treatment.[4] DTD therefore can be considered in addition to the potential impact adverse drug reactions could have on an individual's utility and/ or the out-of-pocket costs (financial burden) medicines may incur. To date, the quantification of DTD in the literature has focused primarily on long-term medicines for cardiovascular disease with values estimated using hypothetical thought experiments such as 'time trade-off' (TTO) exercises where respondents are often asked whether they would forego some length of life not to undergo treatment.[5–7] Estimates from this literature suggest that a substantial proportion of people would be willing to trade some length of life or risk of death not to take a long-term preventative treatment.[5–7]

A previous literature review in 2015[4] highlighted the small number of published cost-effectiveness analyses (CEA) that have included DTD values. Where DTD was incorporated in analysis, it was an important factor impacting on cost-effectiveness results making primary preventative treatments, such as statins, much less likely to be cost-effective.[8–14] In these analyses, DTD values were often assumed[10 13 14] rather than based on empirical estimates. Meanwhile, the small number of previous empirical DTD studies, likely for reasons of practicality, have either adopted small study sizes[5] or sampling frames which are either not representative of patients[7] or the general population.[6] More recently, further CEA[15–18] and other decision-making approaches[19 20] have incorporated DTD values, and found them to be highly influential on results, yet have had to rely on DTD values sourced from the same limited set of empirical studies.

The aim of this study was to build on the previous empirical literature and elicit values of DTD for long-term primary preventative medicines from a mixed sample of the general population and patients. We focus on two populations who might require long-term treatment with medicines: those taking statins for the primary prevention of cardiovascular disease (CVD) and bisphosphonates for the primary prevention of bone fractures. A secondary aim of this study was to explore if survey participant characteristics were associated with the elicited DTD values.

## METHODS

Respondents gave informed consent after reading a participant information sheet online and indicating they agreed to participate in the survey online.

There is no agreement on the appropriate method to elicit DTD.[5–7] This study used cross-sectional surveys, based on a single time point in each survey, to conduct TTO valuation exercises.[21 22] This valuation method is a widely recognised and has been the standard approach for eliciting utility values to generate tariffs used by national decision-making bodies such as the National Institute for Health and Care Excellence (NICE)[23] who have used the EQ-5D-3L to quantify health-related quality of life, valued using a TTO exercise.[24]

### Selection of medicine examples

Statins for primary prevention of CVD was selected because it is an example of an orally administered medicine that is perceived to be benign, but which some people perceive as harmful. Bisphosphonates for the primary prevention of bone fractures was selected because it is an example of a medicine that has an obvious potential impact on day-to-day life. Patients undergoing treatment with bisphosphonates must take the medication on an empty stomach, drink a full glass of water, stand for 30 min after taking the medication and avoid food and drink for a further 2 hours.

### Patient and public involvement

The genesis for the concept of DTD was informed by previous research conducted with patients exploring the impact of medications in those with multimorbidity.[25] To quantify DTD, two patient experts contributed experience of taking medicines long-term, alongside clinical input from the research team, to develop the description of the health (medicine-taking) states used in this study. Separate online surveys were designed for the two selected medicine examples (see online supplemental appendix 1 for the statin survey and online supplemental appendix 2 for the bisphosphonate survey). Pilot studies were conducted for each survey. First, a qualitative pilot study, using the think-aloud method[26 27] from a sample of patients from a General Practitioner (GP) practice in Greater Manchester was used to understand whether the surveys were sufficiently clear for respondents to complete as intended. Second, a quantitative pilot study with 30 respondents (for statin survey and bisphosphonate survey) identified by panel company Dynata was used to allow a preliminary analysis of data collected from the valuation exercise. No changes were made following the quantitative pilot study. The final surveys were formatted and administered online using Sawtooth software.[28] Respondents were sent a secure link to complete a survey.

### Study sample

Two purposive samples of patients taking a statin or bisphosphonate were recruited from two sampling frames: from general practices via the NHS Research Scotland Primary Care Network[29] and the Scottish Health Research Register (SHARE—a register of people living in Scotland allowing recruitment after a search of their medical records).[30] For both, the inclusion criteria for recruiting patients with experience of taking a statin (bisphosphonate) were: prescription of a statin (or a bisphosphonate) in the last year; aged 30 years and over; not been diagnosed with dementia (International Classification of Disease (ICD)-10 code: F00, F01, F02, F03, G10, G20, G30, G31.0, F05.1, R54 and all child codes); not taking a dementia drug (all drugs in British National Formulary chapter 0411); not also taking a bisphosphonate (or statin as appropriate). Patients deemed unsuitable for any reason by their general practitioner were also excluded.

A purposive sample of members of the public were recruited using an online panel company (Dynata).[31] This online panel company provided a sample of respondents with predefined (age 30 years and over; equal gender-balance) characteristics and a demographically-balanced sample from England and Scotland. There were no exclusion criteria for members of the general public who self-select if they were capable of completing an online survey. Respondents from the public or patients could only take part in either a statins or bisphosphonate survey but not both.

Sample-size calculations for valuation exercises are not well established.[32] This study aimed for a pragmatic sample size of a minimum of 500 respondents (250 patients; 250 online panel) for each medicine example. This resulted in a target total sample size of 1000 respondents (500 for the statin survey and 500 for the bisphosphonate survey).

## Valuation exercise

The TTO method employed was similar in approach to that taken by Hutchins *et al*[6 7] when valuing the utility of pill taking. See online supplemental appendix 1 for the whole survey format. The duration within the exercise for daily pill taking was 10 years for statins and 5 years for bisphosphonates. Respondents were asked to imagine taking a pill every day for 10 years (5 years) and then dying. Respondents were then asked whether they would be willing to trade that health state for an alternative where only a single pill was taken at the start of the time period. In this second health state, however, the respondent would live for a shorter duration and then die. A process of 'iteration' was then used to see whether respondents would trade between the health states of differing lengths until a point of indifference occurs between the two health states. The trade-off being quantified here was whether the participant would be willing to choose to live a shorter life but without daily pill taking or alternatively, to have a one-off pill but forego some length of life. The process of iteration to find the potential indifference point followed a standardised process called the Measurement and Valuation of Health protocol whereby a combination of 'bisection' (in which the length of life is always the midpoint of the remaining scale section (bisected)) and 'titration' (in which the length of life is sequentially altered by fixed increments/decrements). The Measurement and Valuation of Health protocol has been recommended previously to encourage comparability between utility values elicited for the purposes of health technology assessment.[32]

The exercises previously described were contextualised with four different background scenarios in order to understand whether DTD values differed depending on the framing of benefits and harms of the medications being used. We asked respondents to consider the exercise with the context that there was: no side effects (scenario 1) associated with any of the pills; mild side effects (scenario 2); severe side effects (scenario 3); and reduced effectiveness (scenario 4). We developed training materials using a storyboard approach[32] to enable the communication of the background concepts for respondents completing each survey (see online supplemental appendices 3 and 4).

## Data analysis

Summary statistics were calculated for a master sample combining data from Scottish Primary Care Research Network and SHARE ('patient respondents') and Dynata ('public respondents') for both statin and bisphosphonate questionnaires. For each of the four questions per respondent, the estimated utility (or the value attached to a health state of daily medicine use) was calculated as the ratio $x/t$. In this calculation, $x$ is the final time period, measured in years, whereby participants were indifferent between living in the health states of Medicine A (one pill taken once) and Medicine B. $t$ represents the health state of a pill taken every day for either statins or bisphosphonates measured as 10 years or 5 years, respectively.[22] The final DTD was calculated by subtracting the estimated utility from 1 or full health. Respondents who indicated they would be unwilling to initiate preventative therapy, by selecting 5 years for Medicine A (or the lowest TTO score attainable of 0.5) in any one of the questions were removed from the data set. This was because we inferred such respondents to have dominant preferences to not undergo preventative treatment.[33 34]

Missing data for background characteristics as well as for the TTO scores were multiply imputed (m=5) using chained equations with predictive mean matching.[35] Differences in respondent TTO scores associated with medicine type (statins vs bisphosphonates), framing of the survey question (question 1, question 2, question 3, question 4), background characteristics (age, ethnicity, sex) and experience of taking medication (pills taken per day, number of times medication taken per day) were explored using ordinary least squares regression accounting for the multiply imputed data sets. Model specification was informed by summary statistics and kernel density plots of the DTD values. Sensitivity to alternative regression models, were explored with competing models compared using root mean squared error. Propensity to trade was explored through a logistic regression for the whole sample with a dummy variable coded 1 for those willing to trade (0.5<TTO<1) and 0 for those unwilling to trade (TTO=1).

For the regression models, a p<0.05 was considered statistically significant. Analysis was carried out in Microsoft Excel and Stata V.16.0 with code available in online supplemental appendix 5.

## RESULTS

Characteristics for statin respondents (n=514) and bisphosphonate respondents (n=365) who were included in the analysis set are reported in table 1. Characteristics for the whole sample (n=1105), including those excluded for having dominant preferences (20.1%) are presented

**Table 1** Description of the sample characteristics

| | Statin survey | | | Bisphosphonate survey | | | |
|---|---|---|---|---|---|---|---|
| | Patient* | Public† | Total | Patient* | Public† | Total | Total |
| | N=227 | N=287 | N=514 | N=86 | N=279 | N=365 | N=879 |
| **Age** | | | | | | | |
| Less than 35 | 1 (0.7%) | 14 (6.1%) | 15 (4.0%) | 0 (0.0%) | 18 (7.9%) | 18 (6.8%) | 33 (5.2%) |
| 35–44 | 4 (2.8%) | 44 (19.1%) | 48 (12.9%) | 1 (2.7%) | 42 (18.5%) | 43 (16.3%) | 91 (14.3%) |
| 45–54 | 6 (4.2%) | 46 (20.0%) | 52 (14.0%) | 2 (5.4%) | 32 (14.1%) | 34 (12.9%) | 86 (13.5%) |
| 55–64 | 43 (30.3%) | 63 (27.4%) | 106 (28.5%) | 10 (27.0%) | 45 (19.8%) | 55 (20.8%) | 161 (25.3%) |
| 65–74 | 66 (46.5%) | 60 (26.1%) | 126 (33.9%) | 15 (40.5%) | 80 (35.2%) | 95 (36.0%) | 221 (34.7%) |
| 75+ | 22 (15.5%) | 3 (1.3%) | 25 (6.7%) | 9 (24.3%) | 10 (4.4%) | 19 (7.2%) | 44 (6.9%) |
| Missing | 85 | 57 | 142 | 49 | 52 | 101 | 243 |
| **Sex** | | | | | | | |
| Female | 49 (34.5%) | 115 (50.0%) | 164 (44.1%) | 33 (89.2%) | 141 (62.4%) | 174 (66.2%) | 338 (53.2%) |
| Male | 93 (65.5%) | 115 (50.0%) | 208 (55.9%) | 4 (10.8%) | 85 (37.6%) | 89 (33.8%) | 297 (46.8%) |
| Missing | 85 | 57 | 142 | 49 | 53 | 102 | 244 |
| **Ethnicity** | | | | | | | |
| White British/Irish | 133 (93.7%) | 211 (91.7%) | 344 (92.5%) | 36 (97.3%) | 203 (89.4%) | 239 (90.5%) | 583 (91.7%) |
| White other | 3 (2.1%) | 10 (4.3%) | 13 (3.5%) | 1 (2.7%) | 8 (3.5%) | 9 (3.4%) | 22 (3.5%) |
| Mixed/multiple ethnic origins | 0 (0.0%) | 1 (0.4%) | 1 (0.3%) | 0 (0.0%) | 5 (2.2%) | 5 (1.9%) | 6 (0.9%) |
| Black/African/Caribbean/black British | 0 (0.0%) | 3 (1.3%) | 3 (0.8%) | 0 (0.0%) | 2 (0.9%) | 2 (0.8%) | 5 (0.8%) |
| Asian/Asian British | 0 (0.0%) | 4 (1.7%) | 4 (1.1%) | 0 (0.0%) | 7 (3.1%) | 7 (2.7%) | 11 (1.7%) |
| Chinese | 0 (0.0%) | 1 (0.4%) | 1 (0.3%) | 0 (0.0%) | 2 (0.9%) | 2 (0.8%) | 3 (0.5%) |
| Other ethnicity | 6 (4.2%) | 0 (0.0%) | 6 (1.6%) | 0 (0%) | 0 (0%) | 0 (0%) | 6 (0.9%) |
| Missing | 85 | 57 | 142 | 49 | 52 | 101 | 243 |
| **Number of pills taken daily** | | | | | | | |
| None | 0 (0.0%) | 91 (39.6%) | 91 (24.5%) | 0 (0.0%) | 77 (33.9%) | 77 (29.2%) | 168 (26.4%) |
| 1 | 4 (2.8%) | 52 (22.6%) | 56 (15.1%) | 4 (10.8%) | 38 (16.7%) | 42 (15.9%) | 98 (15.4%) |
| 2–5 | 104 (73.2%) | 69 (30.0%) | 173 (46.5%) | 23 (62.2%) | 86 (37.9%) | 109 (41.3%) | 282 (44.3%) |
| 6–10 | 31 (21.8%) | 14 (6.1%) | 45 (12.1%) | 5 (13.5%) | 16 (7.0%) | 21 (8.0%) | 66 (10.4%) |
| More than 10 | 3 (2.1%) | 4 (1.7%) | 7 (1.9%) | 5 (13.5%) | 10 (4.4%) | 15 (5.7%) | 22 (3.5%) |
| Missing | 85 | 57 | 142 | 49 | 52 | 101 | 243 |
| **Number of different times pill taken per day** | | | | | | | |
| None | 3 (2.1%) | 94 (40.9%) | 97 (26.1%) | 0 (0.0%) | 75 (33.0%) | 75 (28.4%) | 172 (27.0%) |
| One time per day | 33 (23.2%) | 74 (32.2%) | 107 (28.8%) | 18 (48.6%) | 75 (33.0%) | 93 (35.2%) | 200 (31.4%) |
| Two times a day | 87 (61.3%) | 48 (20.9%) | 135 (36.3%) | 12 (32.4%) | 54 (23.8%) | 66 (25.0%) | 201 (31.6%) |
| Three times a day | 17 (12.0%) | 11 (4.8%) | 28 (7.5%) | 5 (13.5%) | 19 (8.4%) | 24 (9.1%) | 52 (8.2%) |
| More than three times a day | 2 (1.4%) | 3 (1.3%) | 5 (1.3%) | 2 (5.4%) | 4 (1.8%) | 6 (2.3%) | 11 (1.7%) |
| Missing | 85 | 57 | 142 | 49 | 52 | 101 | 243 |
| EQ-5D-3L utility‡ | 0.827 (0.2) | 0.818 (0.2) | 0.822 (0.2) | 0.770 (0.2) | 0.786 (0.2) | 0.783 (0.2) | 0.806 (0.2) |
| Missing | 56 | 41 | 97 | 35 | 33 | 68 | 165 |

*Patient sample was recruited from general practitioners in the NHS Research Scotland Primary Care Network or the Scottish Health Research Register.
†Public sample was recruited from Dynata.
‡Health status measured using the EQ-5D-3L level and transformed into a utility score using Dolan et al.[38]

in online supplemental appendix 6. Demographic characteristics, and experience of taking medicines was not associated with having dominant preferences to avoid preventative treatment (n=226). However, respondents from the public were more likely to have dominant preferences than those with experience of taking medicines (online supplemental appendix 7).

In the analysis set, patients in the statin survey tended to be older versus those in the public cohort with a higher proportion of male respondents (66.5% vs 50.0%). Compared with their public counterparts, patient respondents also tended to take more pills, at more times of the day, yet surprisingly, they also reported a slight improvement in health (EQ-5D-3L utility: 0.827 vs 0.818). Patient respondents in the bisphosphonate sample also tended

to be older than those from the public sample but with a higher proportion of women than in the public cohort (89.2% vs 62.4%). Bisphosphonate patient respondents tended to take more pills, at more times of the day versus those in the public. In contrast to the statin survey, bisphosphonate patient respondents reported comparatively lower health than public respondents (EQ-5D-3L utility: 0.779 vs 0.786, respectively).

Figure 1 shows the pre-imputation distributional properties for TTO values reported by patients and public, stratified by the question-context, for both the statins and bisphosphonates surveys. As can clearly be observed, the spread of responses from participants was highly variable suggesting individual respondents differed greatly in how negatively they valued taking a pill every day. Due

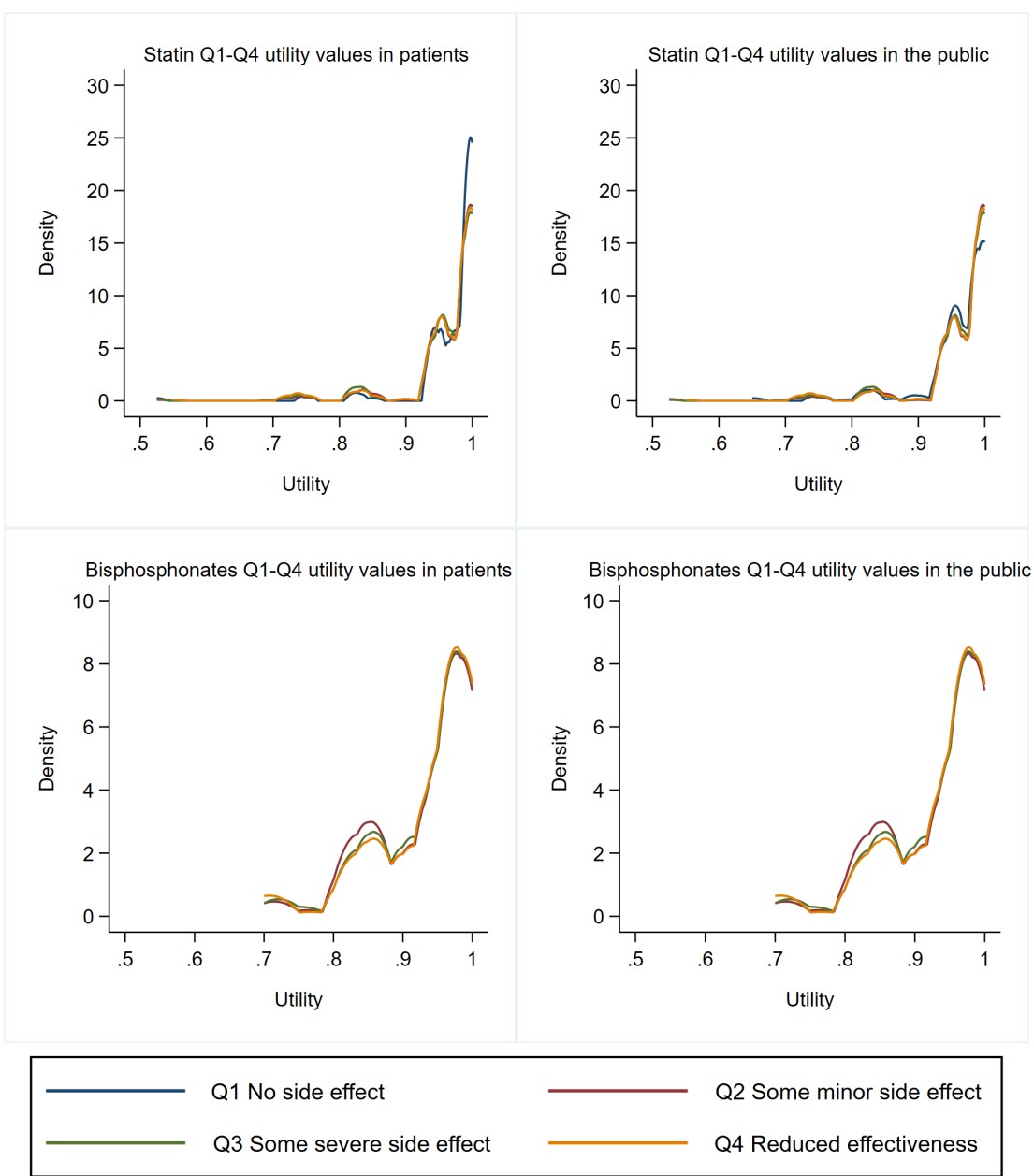

**Figure 1** Kernel density plots showing distribution of time trade-off responses stratified by medicine, question context and respondent type.

**Table 2** Summary statistics of time trade-off values

| Medicine | Respondent | Question context | Mean | SD | Count | p50 | Proportion reporting disutility |
|---|---|---|---|---|---|---|---|
| Statins | Public | No side effects | 0.965 | 0.057 | 237 | 0.983 | 0.746 |
| Statins | Public | Some minor side effects | 0.964 | 0.063 | 233 | 0.995 | 0.721 |
| Statins | Public | Some severe side effects | 0.964 | 0.060 | 232 | 0.988 | 0.725 |
| Statins | Public | Reduced effectiveness | 0.964 | 0.063 | 232 | 0.983 | 0.735 |
| **Statins in the public** | | | 0.964 | 0.061 | 934 | 0.992 | 0.731 |
| Statins | Patients | No side effects | 0.974 | 0.054 | 161 | 0.996 | 0.718 |
| Statins | Patients | Some minor side effects | 0.972 | 0.056 | 156 | 0.996 | 0.718 |
| Statins | Patients | Some severe side effects | 0.968 | 0.060 | 150 | 0.997 | 0.718 |
| Statins | Patients | Reduced effectiveness | 0.970 | 0.055 | 148 | 0.997 | 0.714 |
| **Statins in patients** | | | 0.971 | 0.056 | 615 | 0.997 | 0.717 |
| **Statins all respondents** | | | 0.967 | 0.059 | 1549 | 0.995 | 0.725 |
| Bisphosphonates | Public | No side effects | 0.925 | 0.081 | 219 | 0.967 | 0.860 |
| Bisphosphonates | Public | Some minor side effects | 0.930 | 0.076 | 216 | 0.967 | 0.846 |
| Bisphosphonates | Public | Some severe side effects | 0.932 | 0.076 | 214 | 0.967 | 0.846 |
| Bisphosphonates | Public | Reduced effectiveness | 0.933 | 0.078 | 216 | 0.967 | 0.842 |
| **Bisphosphonates in the public** | | | 0.930 | 0.078 | 865 | 0.967 | 0.849 |
| Bisphosphonates | Patients | No side effects | 0.949 | 0.069 | 42 | 0.983 | 0.814 |
| Bisphosphonates | Patients | Some minor side effects | 0.955 | 0.055 | 40 | 0.967 | 0.826 |
| Bisphosphonates | Patients | Some severe side effects | 0.958 | 0.060 | 40 | 0.975 | 0.802 |
| Bisphosphonates | Patients | Reduced effectiveness | 0.956 | 0.065 | 39 | 0.967 | 0.802 |
| **Bisphosphonates in patients** | | | 0.954 | 0.062 | 161 | 0.967 | 0.811 |
| **Bisphosphonates all respondents** | | | 0.934 | 0.076 | 1026 | 0.967 | 0.840 |
| **All statins and bisphosphonates** | | | 0.954 | 0.068 | 2575 | 0.975 | 0.773 |

p50, Median or 50th percentile.

to this high variability, table 2 presents a wider range of summary statistics than might be typical. Overall, 78% of respondents reported disutility from a taking a pill with a higher proportion indicating some disutility for bisphosphonates (84%) than for statins (73%). Bisphosphonate patients reported much higher mean TTO scores than public respondents (difference: 0.024 (SE: 0.006)) with this difference being statistically significant. For both statins and bisphosphonates, changing the question context did not alter the mean TTO scores by more than 0.01. Irrespective of the type of question or respondent, there was a clear difference between statins and bisphosphonates survey results. Consequently, regression models were run separately on these two cohorts with ordinary least squares regression favoured as it typically produced the least amount of error (online supplemental appendix 8).

The mean conditional TTO value associated with statin use was 0.966 which generates a DTD of 0.034 (calculated as 1, representing full health, minus the estimated utility score). The mean conditional TTO value associated with bisphosphonate use was 0.933 consequently giving a DTD of 0.067. In the statin cohort, none of the explanatory variables were significantly associated with TTO results, with the exception that respondents from the Asian/Asian British group reported higher DTD associated

with statin-use (0.044 greater than white British/Irish respondents) although this finding is only based on a very small sample (n=7) (table 3). In the bisphosphonate cohort, the public provided a DTD level that was 0.011 larger than patients undergoing treatment with bisphosphonates. Furthermore, those who had experience of taking medications more than three times a day provided a much lower DTD value than those who had no experience of daily medicine use. None of the other explanatory variables had a statistically significant impact on DTD size, including the number of pills taken per day. Results from the logistic regression model also reported in table 3 suggest that respondents in the statin survey were less likely to trade than those in the bisphosphonate survey; this is consistent with the finding of higher mean TTO (lower mean DTD) values in the statin scenarios. No other variables in the logistic regression were statistically significant.

## DISCUSSION
In this study we find that long-term statin use is associated with a DTD of 0.034 among people willing to take statins. We find that bisphosphonate use is associated with a DTD of 0.067 among people willing to take bisphosphonates. These values imply that, even if medicines have no

**Table 3** Regression results

| | Statins | Bisphosphonates | Statins and bisphosphonates |
|---|---|---|---|
| | OLS | OLS | Logistic |
| | Utility score (95% CI) | Utility score (95% CI) | Traders† (95% CI) |
| **Sex** | | | |
| Female (reference) | 0 | 0 | 1 |
| Male | −0.00282 (−0.0104 to 0.00478) | 0.00141 (−0.00984 to 0.0127) | 0.859 (0.726 to 1.016) |
| **Ethnicity** | | | |
| White British/Irish (reference) | 0 | 0 | 1 |
| White other | −0.00042 (−0.0157 to 0.0149) | 0.0223 (−0.00800 to 0.0525) | 1.629 (0.873 to 3.04) |
| Mixed/multiple ethnic origins | −0.0105 (−0.0707 to 0.0497) | −0.00997 (−0.0458 to 0.0259) | 1.511 (0.482 to 4.734) |
| Black/African/Caribbean/black British | −0.0349 (−0.0852 to 0.0154) | −0.0128 (−0.0650 to 0.0395) | 1.866 (0.578 to 6.027) |
| Asian/Asian British | −0.0467** (−0.0790 to −0.0144) | −0.0127 (−0.0483 to 0.0228) | 2.516 (0.866 to 7.309) |
| Chinese | 0.0198 (−0.0359 to 0.0755) | −0.0364 (−0.0895 to 0.0168) | 3.813 (0.448 to 32.477) |
| Other ethnicity | 0.0145 (−0.0106 to 0.0395) | 0.00717 (−0.144 to 0.158) | 0.936 (0.418 to 2.097) |
| **Age** | | | |
| Less than 35 (reference) | 0 | 0 | 1 |
| 35–44 | −0.00906 (−0.0299 to 0.0118) | −0.0198 (−0.0496 to 0.0100) | 1.028 (0.658 to 1.607) |
| 45–54 | −0.00428 (−0.0322 to 0.0237) | −0.000518 (−0.0349 to 0.0339) | 1.516 (0.921 to 2.496) |
| 55–64 | 0.00732 (−0.0145 to 0.0292) | 0.00321 (−0.0240 to 0.0305) | 0.943 (0.615 to 1.446) |
| 65–74 | 0.00472 (−0.0176 to 0.0271) | −0.00585 (−0.0352 to 0.0235) | 1.129 (0.711 to 1.792) |
| 75+ | 0.00398 (−0.0237 to 0.0317) | −0.0184 (−0.0545 to 0.0177) | 0.77 (0.424 to 1.397) |
| **Number of pills taken per day** | | | |
| None (reference) | 0 | 0 | 1 |
| 1 | 0.00379 (−0.0120 to 0.0196) | −0.00288 (−0.0216 to 0.0158) | 0.903 (0.658 to 1.239) |
| 2–5 | −0.00776 (−0.0286 to 0.0131) | −0.00873 (−0.0272 to 0.00977) | 1.085 (0.811 to 1.452) |
| 6–10 | −0.00794 (−0.0346 to 0.0188) | −0.00263 (−0.0253 to 0.0201) | 0.867 (0.583 to 1.289) |
| More than 10 | 0.00295 (−0.0308 to 0.0367) | −0.0127 (−0.0411 to 0.0157) | 1.242 (0.63 to 2.446) |
| **Number of different times pill are taken per day** | | | |
| None (reference) | 0 | 0 | 1 |
| One time per day | 0.00475 (−0.0149 to 0.0244) | 0.00892 (−0.00798 to 0.0258) | 0.788 (0.596 to 1.042) |
| Two times a day | 0.0176 (−0.00781 to 0.0430) | 0.00988 (−0.0111 to 0.0309) | 0.757 (0.568 to 1.009) |
| Three times a day | 0.0197 (−0.00677 to 0.0462) | 0.00321 (−0.0254 to 0.0318) | 0.856 (0.562 to 1.303) |
| More than three times a day | 0.0257 (−0.0166 to 0.0680) | 0.0426* (0.00658 to 0.0785) | 1.327 (0.549 to 3.205) |
| **Respondent** | | | |
| Patient (reference) | 0 | 0 | 1 |
| Public | −0.00176 (−0.0118 to 0.00830) | −0.0106* (−0.0202 to −0.000937) | 1.004 (0.831 to 1.214) |
| **Question context** | | | |
| Question 1 (reference) | 0 | 0 | 1 |
| Question 2 | −0.00118 (−0.00978 to 0.00741) | 0.00795 (−0.00695 to 0.0229) | 0.938 (0.746 to 1.178) |
| Question 3 | −0.00266 (−0.0143 to 0.00893) | 0.0065 (−0.00796 to 0.0210) | 0.932 (0.742 to 1.171) |
| Question 4 | −0.00216 (−0.0117 to 0.00737) | 0.00709 (−0.00909 to 0.0233) | 0.941 (0.749 to 1.182) |
| **Sample** | | | |
| Statins | | | 0.531*** (0.444 to 0.635) |
| Constant | 0.969*** (0.947 to 0.991) | 0.941*** (0.907 to 0.975) | 6.11*** (3.615 to 10.335) |
| Observations | 2056 | 1460 | 3516 |
| Individuals | 514 | 365 | 879 |

*p<0.05, **p<0.01, ***p<0.001.
†Those with TTO utility scores <0 are coded. Results represent unstandardised coefficients and 95% CI for the OLS models and adjusted OR and 95% CI for the logistic model coefficients reflect ORs. Score above 1 implies more likely to provide a DTD value or willing to trade, less than 1 implies less likely to provide a DTD.
DTD, direct treatment disutility; OLS, ordinary least squares; TTO, time trade-off.

adverse effects, the act of taking them has a non-trivial impact on people's health-related quality of life. For statins, our study suggests that respondents on average would trade approximately 17 weeks of full health over 10 years while for bisphosphonates it would be more than half a year of life over 10 years. The findings for statins are particularly striking given these treatments are often thought by medical professionals to have minimal impact on users' daily routines.

Existing empirical studies have estimated a range of values of DTD but the general order of the size of the disutility is around 0.01 on average, which is equivalent to a loss of 5 weeks of perfect health over 10 years. In line with previous empirical studies, we find evidence for three different groups or types of respondent: (1) some never trading, suggesting zero disutility associated with treatments; (2) some suggesting they would be unlikely to initiate treatment and (3) some willing to trade length of life for no ongoing treatment, suggesting a DTD. In our survey, the groups willing to trade and generate a DTD made up the majority of those surveyed with approximately 73% and 84% for statins and bisphosphonates respondents, respectively.

We find that estimated mean DTD do not differ depending on whether the treatments were framed as more or less effective or having more or fewer side effects or based on demographic characteristics such as age or sex. Similar to Hutchins *et al*,[6] we do find evidence suggesting that those from a non-white background, in our case an Asian/Asian British ethnic minority background, might perceive a higher level of disutility associated with long-term statin use although this is based on a small sample of participants. This may also be an important part of ethnic health disparities to medication adherence and intensification of treatment. For example, British South Asians have been shown to more slowly intensify diabetes treatment than white groups.[36] We found no difference between patient and public disutilities for statins but we did find that bisphosphonate patients generated smaller disutility values than the values coming from the general public. This finding could support theories rooted in experience utility[37 38] whereby those possessing the 'lived' knowledge of disease, or treatment of disease, do not perceive the negative effects to be as severe as those in the general public, trying to imagine it. Our findings suggest there could be 'hedonistic adaption',[39] with the additional disutility of bisphosphonates being less severe for those who are already taking medicines more than three times a day anyway.

The implications of our findings for future cost-utility analyses evaluating treatment pathways featuring statins or bisphosphonates (and potentially other oral medicines) are not straightforward. On the one hand, CEA should ideally capture the impact of all relevant costs and consequences associated with alternative forms of treatment,[40] so it must be relevant that we have demonstrated that the average person anticipates the act of taking statins or bisphosphonates will have a non-trivial impact

on their health-related quality of life. Accounting for this disutility is likely to reduce the desirability of treatments that are currently considered very cost-effective: estimates of cost-effectiveness for long-term preventative interventions have been shown to be particularly sensitive to the inclusion of DTD.[9–14 41] Indeed, we have previously shown that, for some people for whom guidelines currently recommend statins (eg, those at a 10% risk of a cardiovascular event over 10 years), a DTD that appears moderate in light of the current study (0.015) would result in treatment doing more harm than good.[25 42] Another reason routinely to account for DTD is that, without it, it is not possible to value innovations with positive process characteristics.

On the other hand, the apparent existence of distinct preference groups among our respondents requires careful consideration. A substantial minority of participants repeatedly indicated that they would be unwilling to trade any life expectancy to avoid taking these medicines, suggesting they consider any inconvenience with which they are associated negligible. It would be difficult to deny access to a treatment on the grounds that the average person would be bothered by its process characteristics, which is a danger if population-level cost-effectiveness estimates routinely incorporate average DTD. In view of these conflicting considerations, we recommend that decision-makers review scenarios with and without DTD. If evidence suggests that including DTD would materially alter the balance of benefits, harms and costs associated with treatment, this should be highlighted in population-level guidance, enabling prescribers at an individual level to engage in shared decision-making that gives appropriate weight to the person's preferences for avoiding the treatment's process characteristics. Such an approach fits well with the guideline development methods for NICE,[43] which encourage the explicit identification of 'preference-sensitive decision-points', taking the practicalities of possible treatments into account. Future research could seek to develop tools which could quickly determine the level of DTD which could inform shared decision-making for preference-sensitive decisions.

Our study has several strengths. First, the TTO methods we employ are consistent with the Measurement and Valuation of Health protocol first used to derive the EQ-5D-3L. Moreover, unlike previous studies that have attempted to elicit these values, we use a large representative sample of both public and patients for our estimated values. Finally, due to the challenges associated with understanding and communicating risk, particularly in vulnerable older age-groups, we extensively trialled and developed the use of innovative approaches. For example, we used storyboards and visual arrays to contextualise the TTO exercise and communicate absolute risk. There are some limitations to our study which need to be considered. First, while we made best endeavours to communicate the exercise, the underlying absolute risks and the context for the research question, this had a set of clear trade-offs for participants. Second, the length of the survey, the cognitive burden

and the time required were noted as challenges. Third, some of our respondents reported that they had difficulty understanding the TTO questions while others reported inconsistent values across the survey questions or had missing values, although where there was missingness we did multiply impute, assuming missing at random. Finally, those who took part in our survey ultimately self-selected and this could be related to their ability to complete the survey as well as their self-reported health. Applicability for a different patient or general population should be made based on a careful judgement of the self-reported characteristics summarised for those reporting values in this study cohort.

## CONCLUSION

Long-term preventative interventions, such as statins for CVD or bisphosphonates for bone fractures, have a quantifiable DTD associated with their use. The majority of respondents in our surveys, including public and patient-users, indicate at least some DTD. Bisphosphonates had larger disutilities than statins. Disutility was largely unaffected by respondents' self-reported characteristics but patient users of bisphosphonates did provide smaller DTDs than the public. Future model-based studies assessing the cost-effectiveness of long-term preventative interventions should incorporate DTD values within scenario analyses.

**Acknowledgements** We dedicate the article to the memory of Graham Bell who helped us greatly when formulating the research question and study design. We also wish to thank our other patient and public representatives who were integral to the design of this study. Our sincere thanks goes to all the respondents who piloted the initial survey and completed the final surveys for this study. We acknowledge the insightful contributions of Dr Shona Livingstone, Research Fellow in Statistics, to the survey design.

**Contributors** All authors meet International Committee of Medical Journal Editors (ICMJE) criteria for authorship. AT obtained funding for this study, was involved in formulating the research question, provided advice on the design for the overall study, analysed data and produced a first draft of the manuscript. AT acts as guarantor for this work. J-HY was involved in formulating the research question, completed the process for ethical approval, produced the online version of the survey and contributed to writing the manuscript. BG obtained funding for this study, was involved in formulating the research question, provided advice on the design for the overall study, contributed to the design of the online survey and writing the manuscript. RH helped clean and analyse the data and contributed to writing the manuscript. PD obtained funding for this study, was involved in formulating the research question, provided advice on the design for the overall study, contributed to the design of the online survey and writing the manuscript. GR contributed to data analysis and contributed to writing the manuscript. DM obtained funding for this study, was involved in formulating the research question, provided advice on the design for the overall study and contributed to writing the manuscript. KP obtained funding for this study, formulated the research question, provided advice on the design for the overall study, contributed to the design of the online survey and oversaw data collection and analysis and contributed to writing the manuscript. This manuscript has been read and approved by all the authors.

**Funding** This study/project is funded by the National Institute for Health Research (NIHR) Health Services and Delivery Research Programme (project reference 15/12/22). The views expressed are those of the authors and not necessarily those of the NIHR or the Department of Health and Social Care. The authors had full and sole access to the data, and the funder had no role in the conduct of the research or the decision to publish.

**Competing interests** DM reports that he is supported by a Wellcome Trust Clinical Research Fellowship (214588/Z/18/Z) and is a member of the European Medicines Agency Pharmacovigilance Risk Assessment Committee. No other authors report any conflicts of interest directly relevant to the content of this article.

**Patient and public involvement** Patients and/or the public were involved in the design, or conduct, or reporting, or dissemination plans of this research. Refer to the Methods section for further details.

**Patient consent for publication** Consent obtained directly from patient(s).

**Ethics approval** This study involves human participants and was approved by NHS Health Research Authority Research Ethics Committee (REC reference: 17/NW/0124; project number: 220492). Participants gave informed consent to participate in the study before taking part.

**Provenance and peer review** Not commissioned; externally peer reviewed.

**Data availability statement** No data are available. It is not possible to share the original data as this was a criterion for consent of participants.

**ORCID iDs**
Alexander Thompson http://orcid.org/0000-0003-4930-5107
Bruce Guthrie http://orcid.org/0000-0003-4191-4880
Robert Hainsworth http://orcid.org/0000-0002-3475-800X
Peter Donnan http://orcid.org/0000-0001-7828-0610
Gabriel Rogers http://orcid.org/0000-0001-9339-7374
Katherine Payne http://orcid.org/0000-0002-3938-4350

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
