## [Reviewer comments · BMJ Open]

ARTICLE DETAILS

TITLE (PROVISIONAL)	Quantifying the impact of taking medicines for primary prevention: a time-trade off study to elicit direct treatment disutility in the UK
AUTHORS	Thompson, Alexander; Youn, Ji-Hee; Guthrie, Bruce; Hainsworth, Robert; Donnan, Peter; Rogers, Gabriel; Morales, Daniel; Payne, Katherine

VERSION 1 – REVIEW

REVIEWER	Liu, Chaojie La Trobe University, Public Health
REVIEW RETURNED	11-Nov-2022

GENERAL COMMENTS	Thanks for inviting me to review the manuscript entitled “Quantifying the impact of taking medicines for primary prevention: a time-trade off study to elicit direct treatment disutility”. Overall, the study was well designed and the manuscript was written nicely. Please find below my specific comments: Abstract Too many keywords Introduction Overall, the introduction section was concise and well written. Public or patient choice is often shaped by system and cultural factors. Readers will want to know the situation of DTD globally and in the UK. Is there any relevant literature? A literature review will help justify the rationale of this study. Methods Please clarify whether respondents selected from the Dynata took statin or bisphosphonate. How long did it take for each survey respondent to complete a survey? What is SPCRN? What is “the final selected time period”? What does the utility score indicate? Please describe how DTD values were calculated and what they imply. What “simple regression modelling” was performed: OLS linear? Results How many (percentage) questionnaires were removed from data analyses? Who were more likely to have dominant preferences? Table 1: high numbers of missing values: why? The numbers of respondents with missing values across various characteristics are very similar: does it mean that the same respondents did not respond to all of these questions? Table 3: please specify what are presented in the Table: unstandardised coefficients and 95% CI for the OLS models and adjusted odds ratio and 95% CI for the logistic model?
--

	Discussion “for bisphosphonates it would be more than half a year of life over 10 years”: 10 or 5 years? The discussion section should not present any new findings. “In our survey, the groups willing to trade and generate a DTD made up the majority of those surveyed with approximately 72% and 84% for statins and bisphosphonates respondents, respectively” – was this reported in the results section? Please discuss the implications of the large numbers of missing values and the imputing strategy. Conclusion Please keep consistency in describing the nature of this study: this manuscript has used “cross-sectional”, “case studies” and “experiments”. Which one is correct? Please use the British or US spelling consistently.
--	--

REVIEWER	Green, James University of Limerick Faculty of Education and Health Sciences, School of Allied Health
REVIEW RETURNED	18-Dec-2022

GENERAL COMMENTS	This study addresses a really important question around the negative impacts of taking long-term preventative medications. I have a few recommendations below to improve clarity, but just want to particularly complement the quality of the piloting, both qualitative and quantitative. I do wonder about the ecological validity of a one-off pill; perhaps in future studies a course of several weeks might make a more realistic comparison? However, statins and bisphosphonates good examples of long-term medicines. It would have been useful to include your data analysis code, either as a supplementary appendix, or on a repository. Studies show that “on request” often means it is not available, and becomes less available with time. I requested the code from editor three times over a month, but without a response. This would have led to my review being completed over a month earlier. The manuscript would benefit from a clear statement of experimental design. Seems like it was within/repeated, but spell out clearly rather than requiring the reader to infer. (And eg, when the mention of 10 years or 5 years, are these separate conditions within/between?). And also, for example, the framing of the survey question. These also don’t appear to feature in the discussion. In the introduction, it could be worth discussing related literatures, eg measures of focus on future outcomes for long-term term adherence? Measures of time preference/delay discounting etc. Spell out most abbreviations, as they will not be familiar to the average reader (TTO, CEA, DTD). I note that specifically in the instance of DTD, that spelling it out doesn’t make it obviously easier to read, in that “direct treatment disutility” is a mouthful. Perhaps it would be better to refer to it as “preferring not to take a medicine”, but explain that where that phrase is used, in a technical sense, you mean “direct treatment disutility”? Technical comments:  1. Report descriptive statistics of completion time for participants
---

	2. Report how many excluded for not wanting to initiate preventative therapy. This seems like it could be Supplementary Appendix 5 (but this is not referred to in the text; and then while the label for Appendix 5 says information on excluded participants; that does not seem to be what is in that Appendix. Also, the table there features a and b superscripts not accompanied by notes). 3. Describe ordering process towards indifference point. 4. Table 2 — describe the TTO values presented here in lay terms, so that this table can be interpreted in a standalone fashion 5. It's great being able to see how the survey was laid out. It might just be the review proofs, but these files seem unwieldy. 6. P4 L55 “non-ideal study sizes” — better to say underpowered, or “too small”; be specific about how they are non-ideal 7. P9 L39 — .827 and 0.818 doesn't seem “comparatively better”, but more or less the same?
--	--

VERSION 1 – AUTHOR RESPONSE

Reviewer 1	
1) Too many keywords	We have reduced the number of key words
2) Public or patient choice is often shaped by system and cultural factors. Readers will want to know the situation of DTD globally and in the UK. Is there any relevant literature? A literature review will help justify the rationale of this study.	We agree with the reviewers regarding a literature review and we have previously conducted one to justify the need for this piece of research. We have made that clear at the start of paragraph 3: “A previous review [4] has highlighted the small number of published cost-effectiveness analyses (CEA) that have included DTD values. Where these have been incorporated in analysis, DTD has been an important factor for influencing cost-effectiveness results [8–14]. However, DTD values used have often been assumed [9, 12, 13] rather than based on empirical estimates. Previous empirical DTD studies, likely for reasons of practicality, have adopted non-ideal study sizes [5] or sampling frames which are either not representative of the general population[6] or patients[7], potentially limiting use as part of CEA.”
3) Methods. Please clarify whether respondents selected from the Dynata took statin or bisphosphonate.	We have added the following: “There were no exclusion criteria for members of the general public who self-select if they were capable of completing an online survey. Respondents however could only take part in

	either a statins or bisphosphonates survey but not both.”
4) How long did it take for each survey respondent to complete a survey?	Our pilot studies with face to face interviews suggested 30-60 minutes, after going through the reading material. However, completion times online were not recorded as these values are often troubled because of the way individuals complete surveys online (going away, coming back later etc.)
5) What is SPCRN?	Thank you. We've added the description to the text: “Scottish Primary Care Research Network”
6) What is “the final selected time period”?	Thank you. We have reworded this section to improve clarity: “For each of the four questions per respondent, the estimated utility (or the value attached to a health state of daily medicine use) was calculated as the ratio x/t. In this calculation, x is the final time period, measured in years, whereby participants were indifferent between living in the health states of Medicine A (one pill taken once) and Medicine B. t represents the health state of a pill taken every day for either statins or bisphosphonates measured as 10 years or 5 years respectively [16]. The final DTD was calculated by subtracting the estimated utility from 1 or full health.”
7) What does the utility score indicate?	As above.
8) Please describe how DTD values were calculated and what they imply.	As above. Commentary about what they imply is contained within the discussion: “In this study we find that long-term statin use is associated with a direct treatment disutility (DTD) of 0.034 among people willing to take statins. We find that bisphosphonate use is associated with a DTD of 0.067 among people willing to take bisphosphonates. These values imply that, even if medicines have no adverse effects, the act of taking them has a nontrivial

	impact on people's health-related quality of life. For statins, our study suggests that respondents on average would trade approximately 17 weeks of full health over 10 years whilst for bisphosphonates it would be more than half a year of life over 10 years. The findings for statins are particularly striking given these treatments are often thought by medical professionals to have minimal impact on users' daily routines."
9) What "simple regression modelling" was performed: OLS linear?	Changed to "ordinary least squares regression".
10) Results: How many (percentage) questionnaires were removed from data analyses?	We have added the following to the results section: Characteristics for the whole sample (n=1105), including those excluded for having dominant preferences (20.1%) are presented in Supplementary appendix 5.
11) Results: Who were more likely to have dominant preferences?	We described who were more likely to have dominant preferences in the results section: "Demographic characteristics, and experience of taking medicines was not associated with having dominant preferences to avoid preventative treatment (n=226). However, respondents from the public were more likely to have dominant preferences than those from the public (Supplementary appendix 6)."
12) Results: Table 1: high numbers of missing values: why? The numbers of respondents with missing values across various characteristics are very similar: does it mean that the same respondents did not respond to all of these questions?	Missingness for respondents was approximately 27%. We did not force respondents to give their characteristic information as it may have impacted upon completion rates. Yes, those who did not complete on one part of the survey on their characteristics were more likely not to complete another part on their characteristics.
13) Results: Table 3: please specify what are presented in the Table: unstandardised coefficients and 95% CI for the OLS models and adjusted odds ratio and 95% CI for the logistic model?	We have added the suggested text as a footnote to the table.
14) Discussion: "for bisphosphonates it would be more than half a year of life over 10 years": 10 or 5 years?	For 10 years. We do not think adding text here to clarify is necessary.

15) Discussion: The discussion section should not present any new findings. “In our survey, the groups willing to trade and generate a DTD made up the majority of those surveyed with approximately 72% and 84% for statins and bisphosphonates respondents, respectively” – was this reported in the results section?	Thank you for highlighting this. We have added the following to the results: “Table 2 and Error! Reference source not found. show the pre-imputation distributional properties for TTO values reported by patients and public, stratified by the question-context, for both the statins and bisphosphonates surveys. Overall, 78% of respondents reported disutility from a taking a pill with a higher proportion indicating some disutility for bisphosphonates (84%) than for statins (73%).” We have rounded 72.5% to 73% and corrected this in the text.
16) Discussion: Please discuss the implications of the large numbers of missing values and the imputing strategy.	We do not necessarily agree that there were a large number of missing values. Around 25-30% of patient characteristic variables were missing but these were multiply imputed. We have added the following to the discussion: “Third, some of our respondents reported that they had difficulty understanding the time trade-off questions whilst others reported inconsistent values across the survey questions or had missing values, although where there was missingness we did multiply impute, assuming missing at random. Finally, those who took part in our survey ultimately self-selected and this could be related to their ability to complete the survey as well as their self-reported health. Applicability for a different patient or general population should be made based upon a careful judgement of the self-reported characteristics summarised for those reporting values in this study cohort.”
17) Conclusion: Please keep consistency in describing the nature of this study: this manuscript has used “cross-sectional”, “case studies” and “experiments”. Which one is correct?	We have removed reference to case studies (replaced with ‘medicine examples’) and experiments (exercise) as these terms could cause misconceptions. We have also added the following to the start of the methods section: “This study used cross sectional surveys, based on a single time point in each survey, to conduct time trade off valuation exercises [15] [16]”

18) Conclusion: Please use the British or US spelling consistently.	Thank you for highlighting this. We have checked for US spellings but cannot find any. However, as is the case with these sorts of things, a different set of eyes might be able to identify what we have missed.
Reviewer 2	
1) I do wonder about the ecological validity of a one-off pill; perhaps in future studies a course of several weeks might make a more realistic comparison? However, statins and bisphosphonates good examples of long-term medicines.	Thank you for the suggestion. Whilst recognising few clinical examples, we took advice from our pilot study and expert patients that our approach would be the most meaningful way to setup the question we were interested in answering.
2) It would have been useful to include your data analysis code, either as a supplementary appendix, or on a repository.	Thank you. We have included a script file for the analysis and outputs.
3) The manuscript would benefit from a clear statement of experimental design. Seems like it was within/repeated, but spell out clearly rather than requiring the reader to infer. (And eg, when the mention of 10 years or 5 years, are these separate conditions within/between?).	Thank you for identifying this limitation with our reporting in the paper. We have added the following to make the experimental approach more explicit: “This study used cross sectional surveys, based on a single time point in each survey, to conduct time trade off valuation exercises [15] [16].” Please also see the rest of our response to Reviewer 1, point 17. For the time periods, we have altered the description to make clearer which time period goes with which medicine: “For each of the four questions per respondent, the estimated utility (or the value attached to a health state of daily medicine use) was calculated as the ratio x/t. In this calculation, x is the final time period, measured in years, whereby participants were indifferent between living in the health states of Medicine A (one pill taken once) and Medicine B. t represents the health state of a pill taken every day for either statins or bisphosphonates measured as 10 years or 5 years respectively [16]. The final DTD

	was calculated by subtracting the estimated utility from 1 or full health.” We have also added the following in the study sample section: “Respondents from the public or patients could only take part in either a statins or bisphosphonate survey but not both.”
4) In the introduction, it could be worth discussing related literatures, eg measures of focus on future outcomes for long-term adherence? Measures of time preference/delay discounting etc.	Whilst we agree that these other literatures and concepts are related and interesting, we think the paper is already quite complex for the reader. We would be reluctant to add further concepts which might detract from the message of this research / approach for capturing disutility.
5) Spell out most abbreviations, as they will not be familiar to the average reader (TTO, CEA, DTD).	We have changed TTO to time trade-off throughout and CEA to cost-effectiveness analysis. However, we wish to keep DTD as we believe it is a reasonable short-hand that we would like to encourage. This term was also used for our previous paper on this topic: Thompson AJ, Payne K, Guthrie B. Do pills have no ills? Capturing the impact of direct treatment disutility. Pharmacoeconomics. 2015;
6) Report descriptive statistics of completion time for participants 2.	As noted to Reviewer 1, we did not record completion times. We have considerable experience within the research group of conducting these sorts of surveys. It is very common for respondents to start answering and then come back later, making digital ‘completion times’ an unfortunately noisy measurement. That is also reflected in the literature.
7) Report how many excluded for not wanting to initiate preventative therapy. This seems like it could be Supplementary Appendix 5 (but this is not referred to in the text; and then while the label for Appendix 5 says information on excluded participants; that does not seem to be what is in that Appendix. Also, the table there features a and b superscripts not accompanied by notes).	We have added the following to the opening of the results section: “A total of 1105 individuals completed the surveys (full sample characteristics in Supplementary Appendix 5). After removing respondents unwilling to initiate preventative therapy on any one of the questions (n=226) a

	total of 879 individuals completed the two surveys.” We have also added the footnotes for the table. Thank you for spotting that.
8) Describe ordering process towards indifference point	We added the following when describing titration towards an indifference point: “The process of iteration to find the potential indifference point followed a standardised process called the Measurement and Valuation of Health protocol whereby a combination of ‘bisection’ (in which the length of life is always the midpoint of the remaining scale section (bisected)) and ‘titration’ (in which the length of life is sequentially altered by fixed increments/ decrements). The Measurement and Valuation of Health protocol has been recommended previously to encourage comparability between utility values elicited for the purposes of health technology assessment [26].”
9) Table 2 — describe the TTO values presented here in lay terms, so that this table can be interpreted in a standalone fashion 5.	We have introduced the following wording into the results section: “Error! Reference source not found. shows the pre-imputation distributional properties for time trade-off values reported by patients and public, stratified by the question-context, for both the statins and bisphosphonates surveys. As can clearly be observed, the spread of responses from participants was highly variable suggesting individual respondents differed greatly in how negatively they valued taking a pill every day. Due to this high variability, Table 2 presents a wider range of summary statistics than might be typical. Overall, 78% of respondents reported disutility from a taking a pill with a higher proportion indicating some disutility for bisphosphonates (84%) than for statins (73%).”
10) It’s great being able to see how the survey was laid out. It might just be the review proofs, but these files seem unwieldy.	Unfortunately, it was the nature of the online survey that any reproduction of it must be created on a large ‘flat’ file as in the Appendix. We were unable to maintain the survey on the server for any time longer than our exercise and so we cannot provide a ‘link’ for it. In future, we will consider other ways to demonstrate the

	survey, perhaps using a video of a respondent clicking through it on screen.
11) P4 L55 “non-ideal study sizes” — better to say underpowered, or “too small”; be specific about how they are non-ideal 7. P9 L39 — .827 and 0.818 doesn’t seem “comparatively better”, but more or less the same?	Thank you. We have changed “non-ideal study sizes” which is ambiguous to “small”, which is more to the point. We slightly disagree regarding the difference in utilities. Given the context of the sizes of disutility the paper is focused on, a difference of 0.827 and 0.818 is non-trivial, particularly given we would have expected older patients to report lower health. However, we have changed “comparatively better” to a “slight improvement”.

VERSION 2 – REVIEW

REVIEWER	Liu, Chaojie La Trobe University, Public Health
REVIEW RETURNED	04-Apr-2023
GENERAL COMMENTS	Strength and limitations: "contextualize" or "contextualise"? Page 10: "However, respondents from the public were more likely to have dominant preferences than those from the public": please correct this sentence. Table 1: please correct the formatting of the footnote.